# Visual Chain-of-Thought: Bridging Logical Gaps with Multimodal Infillings

## Abstract

Recent advances in large language models elicit reasoning in a chain-of-thought that allows models to decompose problems in a human-like fashion. Though this paradigm improves multi-step reasoning ability in language models, it is limited by being unimodal and applied mainly to question-answering tasks. We claim that incorporating visual augmentation into reasoning is essential, especially for complex, imaginative tasks. Consequently, we introduce **VCoT**[1] a novel method that leverages chain-of-thought prompting with vision-language grounding to recursively bridge the logical gaps within sequential data. Our method uses visual guidance to generate synthetic multimodal infillings that add *consistent* and *novel* information to reduce the logical gaps for downstream tasks that can benefit from temporal reasoning, as well as provide interpretability into models' multi-step reasoning. We apply VCoT to the Visual Storytelling and WikiHow summarization datasets and demonstrate through human evaluation that VCoT offers novel and consistent synthetic data augmentation beating chain-of-thought baselines, which can be used to enhance downstream performance.

## 1 Introduction

A recent landmark result in natural language processing is *chain-of-thought prompting* (CoT) (Wei et al., 2022; Kojima et al., 2022), whereby decomposing complex problems into simple steps for input to a large language model (LLM) confers improved performance on a variety of tasks (Lampinen et al., 2022). While CoT demonstrates impressive performance on tasks in the text-only question-answering domain, it is unclear how this technique can be generalized to multimodal (e.g., vision-language) settings.

We hypothesize that one core benefit to CoT in the text domain is that it prompts an LLM to *fill in logical* or *sequential gaps*. With this frame, both the methodology and benefits of extending CoT to the visual domain becomes plausible. Many vision-language (VL) reasoning tasks, such as virtual assistants (Qiu et al., 2021), navigators (Anderson et al., 2018), and decision-makers (Huang et al., 2022), require some degree of sequential data understanding. However, these techniques are currently restricted to "reason" over a limited set of input data (e.g., key frames) which may contain logical gaps, hindering task-specific performance (Figure 1). This

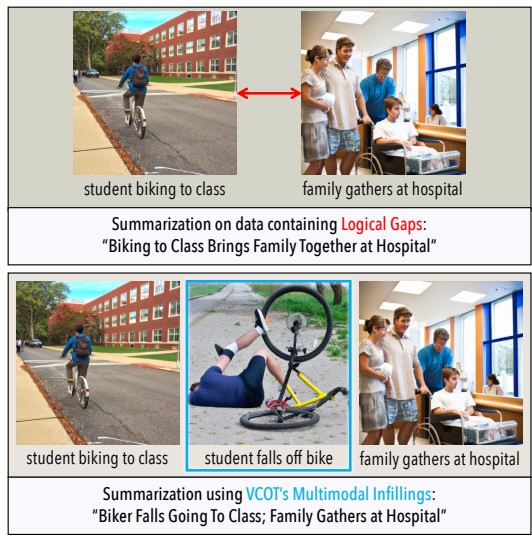

Figure 1: Sequences often contain logical gaps between elements that can limit reasoning tasks; our proposed **Visual Chain-of-Thought** method bridges these gaps with multimodal infillings to downstreaming reasoning.

---

[1]The source code will be made publicly available at camera ready.

leads us to our core idea: *we can extend CoT prompting into the vision and language domain by integrating generative image models to produce intermediate images*.

We argue incorporating the visual modality into CoT can help bridge logical gaps in two ways. First, multi-step reasoning with visuals better fills logical gaps because images capture additional information that unimodal text cannot. Second, visual chains mimic human imagination which creates novel solutions (Tan & Bansal, 2021; Lu et al., 2022; Zhu et al., 2022) and provide interpretability (Wang et al., 2022b) into decision making. One imagined picture provides a thousand-word insight to enhance computer reasoning.

We propose **Visual Chain-of-Thought** (VCoT), which combines the efficiency, robustness, and multi-step reasoning of CoT with the multimodal capabilities of vision-language models. VCoT synthetically augments sequential datasets and bridges logical gaps by recursively generating *multimodal infillings* and using the synthetic data to improve downstream task performance. These synthetic generations also serve as human-interpretable insights into AI systems' ability of multi-step reasoning. We demonstrate that VCoT creates *consistent* and *novel* synthetic data that enhances downstream performance on the VIST (Huang et al., 2016) and WIKIHOW (Koupaee & Wang, 2018) datasets. Our main contributions are:

- We propose Visual Chain-of-Thought for sequential data to generate synthetic text-visual pairs as data augmentation for downstream reasoning.
- We devise a consistency and novelty-driven approach to recursively generate multimodal infillings that augment faithful, relevant context.
- We demonstrate the effectiveness of our method through human evaluation, showing improvements in sequential reasoning.

## 2 RELATED WORK

**Training-free Prompting Paradigm.** Language model pre-training and fine-tuning ushered in an era of higher accuracy, more data and compute-efficient NLP (Devlin et al., 2018; Howard & Ruder, 2018; Gao et al., 2020; Dodge et al., 2020). As models continued to scale, few and zero-shot techniques for many general language tasks have become feasible and performant (Kojima et al., 2022; Brown et al., 2020; Huang et al., 2022). Recent research focuses on optimizing prompts in order to improve the performance of zero-shot techniques, such as in-context inference (Lampinen et al., 2022) and prompt ensembling (Wang et al., 2022a). In light of the impressive performance of prompt-optimization efforts for zero-shot NLP techniques, it becomes natural to ask: *can multimodal vision & language tasks benefit from LLM generalization techniques?* Our work seeks to bridge the best of both modalities through using efficient strategies to reason with LLMs while leveraging visual imagery for sequential tasks.

**Data Augmentation.** Another trend within language modeling is to augment LLMs with external data. These solutions include leveraging dense passage retrieval (Lewis et al., 2020), prompt editing (Madaan et al., 2022), human guidance (Guan et al., 2020), and conversation history in the conversational AI setting (Ghazvininejad et al., 2018). External data provides complementary knowledge to the models' training data, improves robustness without rigorous fine-tuning, and enables systems to be time-agnostic. Previous work studies visual knowledge incorporation for language understanding (Tan & Bansal, 2021; Lu et al., 2022) and generation (Zhu et al., 2022), but it has yet to be applied to sequential reasoning. We are the first propose a vision-augmented CoT, which infills temporal data to provide data augmentation to reduce the logical leaps in sparse data that may limit general sequential generation tasks.

**Intermediate Infillings.** In earlier research relating to synthetic infillings, researchers point out significant lagging performance in temporal reasoning (Vashishtha et al., 2020) and offer advancements (Zhou & Hripcsak, 2007; Dorn, 1992). Works have separately looked to augment sparse data with intermediate logical links (Kotnis et al., 2015) and create intermediate representations between sequential elements (Hong et al., 2023). While promising, these advancements are task-specific, unimodal, and computationally expensive. Our novel application of VCoT draws inspiration from previous research and adds synthetic multimodal logical infillings to strengthen temporal reasoning for general sequential datasets.

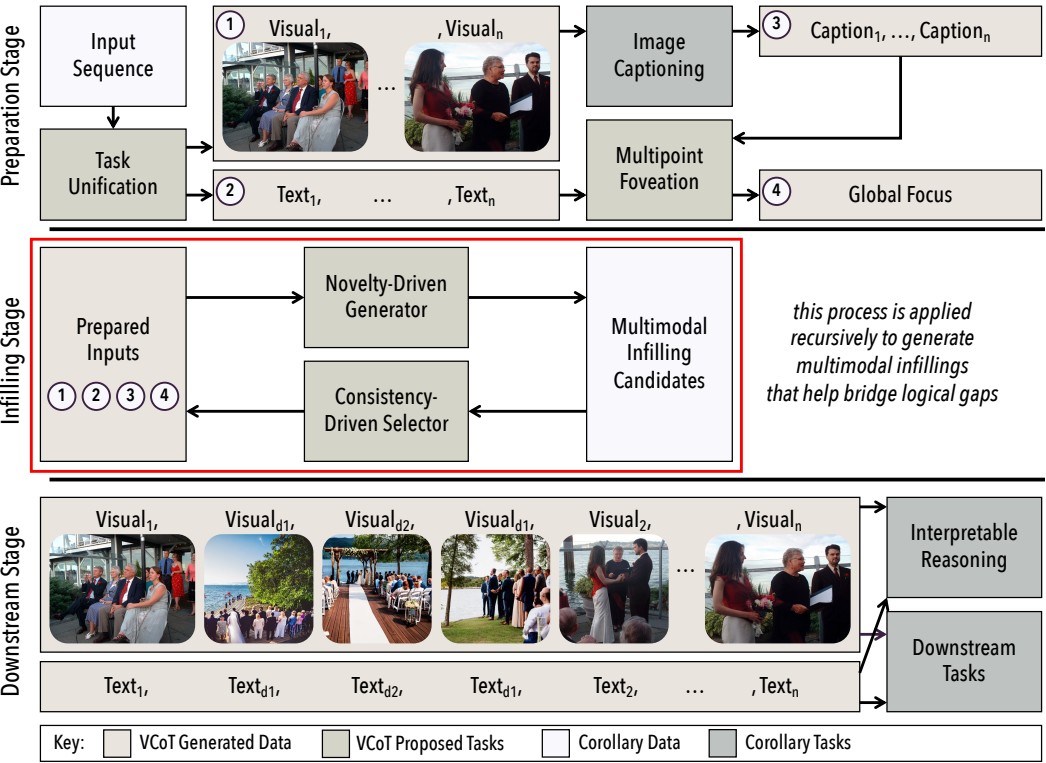

Figure 2: Overview of our novel Visual Chain-of-Thought method. The preparation stage unifies an arbitrary input sequence as a sequence of visual-text pairs (§4.1), constructs associated captions, and a global focus (§4.2). Next, VCoT recursively generates multimodal infillings by first producing novel candidates (§4.3) and then selecting the most consistent option (§4.4). VCoT's multimodal infillings provide interpretability into the reasoning process and synthetic data for downstream tasks.

## 3 PROBLEM FORMULATION

To improve temporal reasoning in language models, we define the **multimodal infilling** task to bridge the logical gaps in sequential data with synthetic multimodal infillings. Given two consecutive text-visual pairs $\{(v_{i-1}, t_{i-1}), (v_{i+1}, t_{i+1})\}$, we generate an infilling $\{(v_i, t_i)\}$ through generator $g$ (e.g., VCoT) using model $M$ (Equation 1). After we generate multiple such infillings, we select the best $(v_i^{best}, t_i^{best})$ among the candidates using judgement function $j((v_i, t_i))$ (e.g., CLIP similarity) (Equation 2). In keeping with the training-free formulation of CoT, we limit ourselves to generating and selecting candidate intermediate steps through prompting existing pretrained models. For a downstream task $T$ (e.g. visual storytelling, instruction summarization) measured by performance $pe$ (e.g., novelty, consistency, coherence, descriptiveness), $(v_i^{best}, t_i^{best})$ is an optimal infilling $(v_i^{opt}, t_i^{opt})$ to be kept if its' addition improves the downstream task performance (Equation 3). Otherwise, the optimal infilling is redundant or damaging, and it is set to `null` (Equation 4). Our VCoT is a method that serves as a generator $g$ and a way to select $(v_i^{best}, t_i^{best})$, but we leave determining optimality of recursive termination to future work.

$$t_i := g(t_{i-1}, t_{i+1}, M) \tag{1}$$

$$t_i^{best} := \arg\max_{t_i}[j(t_i)] \tag{2}$$

$$v_i := g(t_i^{best}, M) \tag{3}$$

$$v_i^{best} := \arg\max_{v_i | t_i^{best}}[g(t_i^{best}, M)] \tag{4}$$

$$pe(T, [(v_{i-1}, t_{i-1}), (v_i^{best}, t_i^{best}), (v_{i+1}, t_{i+1})]) > pe(T, [(v_{i-1}, t_{i-1}), (v_{i+1}, t_{i+1})]) \tag{5}$$

$$(v_i^{opt}, t_i^{opt}) := \begin{cases} (v_i^{best}, t_i^{best}) & \text{Eq. 6 holds} \\ \texttt{null} & \text{otherwise} \end{cases} \tag{6}$$

## 4 VCoT Multimodal Infilling Generation

We propose the VCoT method as a solution to the multimodal infilling task. To generate high-quality multimodal infillings, we use a combination of CoT and vision-language models. We propose the following pipeline[2] (Figure 2):

1. We transform text-only datasets into multimodal text-visual pairs for task unification (§4.1).
2. We identify the **multipoint foveation** to extract the global focus to guide the generation of consistent multimodal infillings (§4.2).
3. We generate our synthetic data through a **novelty-driven recursive infilling** (§4.3) and **consistency-driven visual augmentation** (§4.4) approach to provide interpretability for multi-step reasoning and reduce logical gaps to aid downstream task performance.

### 4.1 Task Unification

To apply VCoT to general sequences, we first reformat sequential data into text-visual pairs. For text-only sequences, we generate candidate visuals $\{v_1', \ldots, v_n'\}$ for the corresponding input text sequence $\{t_1, \ldots, t_n\}$ by using Stable Diffusion. Here, each $v_i'$ is a set of multiple candidate visuals $\{v_{i1}, \ldots, v_{ij}\}$ for each $t_i$. We then assess the similarity of each candidate visual in $v_i'$ to the grounding input text $t_i$ using CLIP embeddings and select the most similar candidate. This yields a sequence of consistent visuals $\{v_1, \ldots, v_n\}$ that form pairs with the input text, unifying general sequences into a series of text-visual pairs.

### 4.2 Multipoint Foveation

To keep the multimodal infillings consistent with the input sequence, we use foveation to identify the overall main focus (Mei et al., 2022). Since pairwise sequential elements may omit relevant and important fixation points, we define **multipoint foveation** (MPF) to identify all of the core fixation points (e.g., setting, characters) of the entire visual-text input sequence (Figure 3). We then project the visual-text pairs into a unimodal text space by captioning the visuals: $\{(v_1, t_1)...(v_n, t_n)\} \rightarrow \{(c_1, t_1)...(c_n, t_n)\}$. The projected output, along with three to four hand-written few-shot exemplars, is fed into GPT-3.5 to generate a maximum likelihood[3] summary from which the MPF $f$ is extracted (Equation 5). The foveation guides infillings to be consistent and not introduce excessive information. We provide a qualitative example showing the effectiveness of MPF (Figure 14).

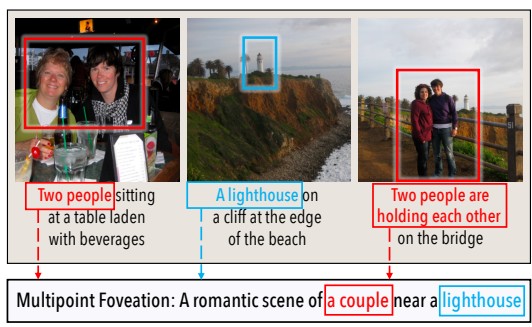

Figure 3: Visualization of our multipoint foveation method, which extracts a global focus from a sequence of text-visual pairs to guide new infilling generations.

$$MPF((c_i, t_i), M) := \arg\max_f (\mathbb{P}(f|(c_i, t_i), M)) \tag{7}$$

### 4.3 Novelty-Driven Recursive Infilling

The judgment function, $j((v_i, t_i))$ (Equation 2), judges the generated multimodal infillings based on two metrics: consistency and novelty. **Consistency** ensures that the *infillings maintain faithful*

---

[2]https://sites.google.com/view/vcot/home
[3]Likelihood is defined in Appendix A.3.1.

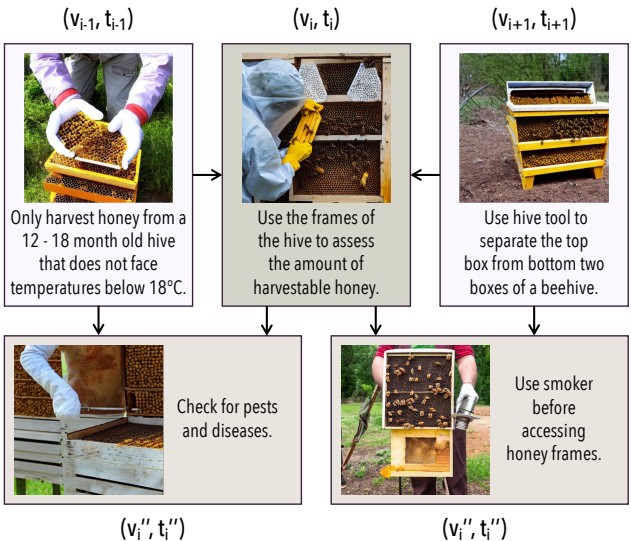

Figure 4: Infilling examples. Given three inputs: image, text pairs: $(v_{i-1}, t_{i-1})$, $(v_{i+1}, t_{i+1})$, and $depth = 2$, VCoT recursively generates $(v_i, t_i)$, $(v_i', t_i')$, and $(v_i'', t_i'')$ that fill in the logical gaps.

*details from the surrounding steps*, while **novelty** ensures that the *infillings add relevant and new information that accurately bridges the logical gaps.*

We infill a set of visual-text pairs $(v_{i-1}, t_{i-1})$ and $(v_{i+1}, t_{i+1})$ using foveation $f$ with new information bridging the logical gap (Appendix A.1, Algorithm 2). We opt for a recursive approach rather than an iterative approach when infilling logical gaps as some information may have more logical gaps than others and may require more infillings. A recursive approach makes it easier to dynamically determine when infillings are not beneficial to the task and when to stop generating them in contrast to an iterative approach. Our approach generates multiple depths of infillings to add valuable new, relevant, and consistent multimodal information for downstream tasks (Figure 4).

### 4.4 CONSISTENCY-DRIVEN VISUAL AUGMENTATION

To explicitly guide consistent infilling generation, we use multipoint foveation (§4.2) to ground our generations to the input sequence and CLIP to select the most consistent recursively generated infilling with respect to their surrounding pair (Algorithm 1). Specifically, we generate five candidate text-infillings with GPT-3.5 and compare them to their surrounding visuals using CLIP embeddings and select the most similar candidate. When generating text-infillings, we prompt GPT-3.5 with information from the surrounding text to provide logical context that maintains consistency, novelty, and sequential order. To generate a consistent, sequential visual, we prompt STABLE DIFFUSION with the selected text-infilling to generate four candidate visuals. Then, we choose the candidate visual most consistent using CLIP embeddings.

To determine the recursive stopping condition, we experiment with both a fixed recursive depth and an learned approach by prompting GPT-3.5 to classify whether a logical gap remains. Empirical results demonstrate the GPT-3.5-halting approach shows inconsistent performance that adds significant noise. Instead, we opt for a fixed depth, $depth$-$limit$=2, which balances sufficiently filling logical gaps with not injecting irrelevant information.

## 5 EXPERIMENTS

We leverage leading vision and language generation models STABLE DIFFUSION and GPT-3.5 for synthetic data augmentation, along with CLIP guidance and OFA captioning. We test our method on the VIST and WIKIHOW datasets due to their sequential composition to show the effectiveness of VCoT in bridging reasoning gaps.

---

**Algorithm 1:** genInfilling(prev, next, foveation)

---

**Data:** prev $(v_{i-1}, t_{i-1})$ and next $(v_{i+1}, t_{i+1})$ infillings and foveation $f$
**Result:** single intermediate infilling $(v_i, t_i)$
STEP 1: GENERATE INTERMEDIATE TEXT
1 Generate captions $c_{i-1}, c_{i+1}$ for visuals $v_{i-1}, v_{i+1}$, respectively
2 Generate five candidate texts $t'_1, \ldots, t'_5$ using GPT-3.5 with input $((c_{i-1}, t_{i-1}), (c_{i+1}, t_{i+1}), f)$
3 Select highest similarity candidate text $t_{best}$ by comparing CLIP embeddings of $t'_1, \ldots, t'_5$ to the CLIP embeddings of $t_{i-1}$ and $t_{i+1}$
STEP 2: GENERATE INTERMEDIATE VISUAL
4 Generate four candidate visuals $v'_1, \ldots, v'_4$ using STABLE DIFFUSION with input $t_{best}$
5 Select highest similarity candidate visual $(v_{best})$ by comparing CLIP embeddings of $v'_1, \ldots, v'_4$ to the CLIP embedding of $t_{best}$
6 **return** $(v_{best}, t_{best})$

---

Figure 5: VCoT generates candidate infillings and uses CLIP embs to select the most consistent one.

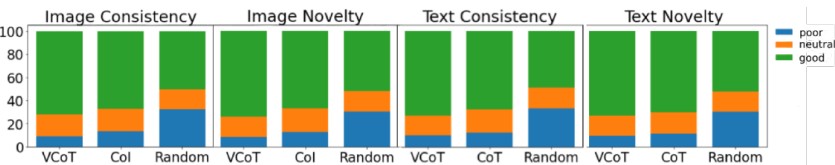

Figure 6: Consistency and novelty rating distributions of VCoT text-visual infillings compared to the Chain-of-Thought (CoT), Chain-of-Images CoI, and random baselines. Here multimodal infillings are selected as "good", "neutral", or "poor", and VCoT again surpasses all other baselines.

Table 1: Consistency and novelty ratings of VCoT intermediate visual-text infillings compared to the Chain-of-Thought (CoT) + Chain-of-Images (CoI) and random baselines, represented as wins-tie-loss percentages. VCoT has higher win percentages compared to both baselines.

| VCoT vs. Baseline | Image Consistency | | | Text Consistency | | | Image Novelty | | | Text Novelty | | |
|---|---|---|---|---|---|---|---|---|---|---|---|---|
| | Win($\uparrow$) | Tie | Lose($\downarrow$) | Win($\uparrow$) | Tie | Lose($\downarrow$) | Win($\uparrow$) | Tie | Lose($\downarrow$) | Win($\uparrow$) | Tie | Lose($\downarrow$) |
| CoT+CoI | 26.82 | 53.02 | 20.16 | 28.07 | 52.21 | 19.73 | 30.13 | 50.24 | 19.63 | 25.77 | 52.86 | 21.37 |
| Random | 30.13 | 50.24 | 19.63 | 43.40 | 39.27 | 17.33 | 43.66 | 38.95 | 17.39 | 41.87 | 40.36 | 17.77 |

## 5.1 EXPERIMENTAL SETTINGS

**Datasets.** We use VIST and WIKIHOW to evaluate the quality of VCoT's synthetic multimodal infillings and their impacts on visual storytelling and instruction summarization, respectively. VIST is a visual storytelling dataset consisting of sequences of five text-visual pairs representing a single story (Huang et al., 2016). The test set contains 2021 stories. The clear gaps between flickr visuals and human-written captions, and pairwise sequential elements often create sizable logical gaps. WIKIHOW is a text summarization dataset containing "How-To" articles, with 6000 test set articles (Koupaee & Wang, 2018). VCoT's synthetic multimodal infillings between logically distanced instructions can decompose difficult instructions. VIST provides us with a standard sequential text-visual dataset, and we showcase WIKIHOW as a text-only dataset for our task unification process (§4.1). For downstream evaluation of WIKIHOW, we input "How-To" articles as a sequence of paragraphs that we seek to summarize into descriptive, human-understandable instructions, a slightly different approach than strict summarization.

**Implementation Details.** VCoT generates five textual infilling candidates using GPT-3.5, one with zero temperature and four with 0.5 temperature. In our current approach, we select the best textual infilling (Equation 2) and input it input into the STABLE DIFFUSION generator. We otherwise use zero temperature to maximize consistency. To evaluate the infillings themselves, we combine WIKIHOW and VIST examples and present 7062 generated infillings whose scores are provided by Amazon Mechanical Turk crowd workers who pass an attention check. For the downstream tasks, we evaluate each dataset separately, considering 227 full WIKIHOW articles and 266 VIST stories.

We use GPT-3.5 (`text-davinci-003`) for all language tasks over open-source alternatives as GPT-3.5 demonstrates stability over open-source alternatives. We utilize an out-of-the-box image captioning checkpoint (`OFA-base`) to show the generality of VCoT and demonstrate performance without the need for task-specific fine-tuning. We use STABLE DIFFUSION (Rombach et al., 2022) (`Stable-Diffusion 1.4`) for image generation. We use CLIP for multimodal similarity comparisons to guide the cross-modal generation.

## 5.2 HUMAN EVALUATION

The reason we opt for human evaluation over automatic evaluations is to account for multi-modality and ensure complex analysis of filling logical gaps with novel and consistent synthetic tokens. Further, we do not seek to compare with the datasets' ground truths because the ground truths in VIST and WIKIHOW often are incoherent or undescriptive. Instead, our data augmentation seeks to surpass ground truth results by synthetically filling logical gaps and providing human-interpretable multi-step reasoning.

**Evaluation Criteria.** For both evaluations, we use the `novelty` and `consistency` metrics defined in §4.3. As additional downstream evaluation metrics, we add coherence for storytelling and descriptiveness for instruction summarization. `Coherence` confirms that the output *flows logically together as an interesting and creative story.* `Descriptiveness` verifies that the *generated summaries describe accurately and with detail the steps of the "How-To" article.* We select our human evaluation criteria based on BARTSCORE (Yuan et al., 2021) and our goal of generating relevant logical infillings.

**Evaluation Details.** We ask human annotators to follow the evaluation criteria (*novelty*, *consistency*, *coherence*, and *descriptiveness*) using win-tie-lose comparison, a common approach in human evaluation (Gu et al., 2021; Yang et al., 2019), which reduces variance and increases inter-annotator agreement. Since our method is the first to generate multimodal infillings, exist-

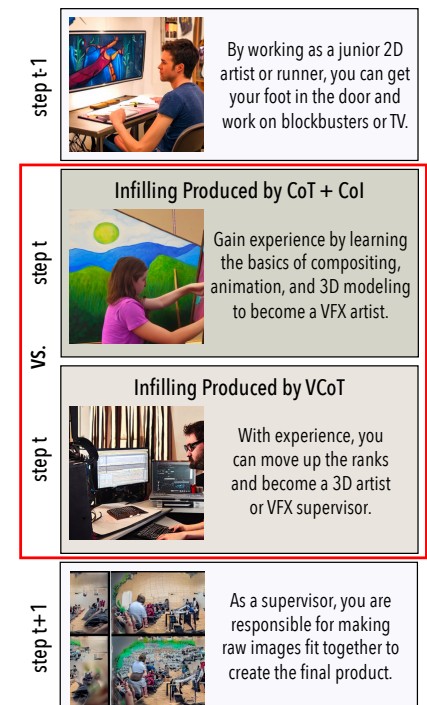

Figure 7: Compared to the Chain-of-Thought (CoT) and Chain-of-Images (CoI) baselines, VCoT infills more consistent and relevant novel information to the input steps on "How to Become a VFX Artist." CoT and CoI generate their respective infilling unimodally whereas VCoT infills using context from both modes.

ing vision-language models are not well-suited as baselines. Instead, we use head-to-head comparisons of VCoT's text-visual pairs with purely textual chain-of-thought (CoT) paired with the purely visual chain-of-images (CoI) performed in parallel, using the same recursion depth. To evaluate the infillings, we additionally compare with a random baseline, which selects a random multimodal pair from our generated examples. To evaluate downstream task performance, we also compare with generation without using generated infillings, as well as with the ground truth of each dataset. We hired a total of 20,534 workers using the Mechanical Turk platform, and paid a rate of $.30/HIT for our labeling tasks to an average hourly wage of $15/hr.

## 5.3 QUALITY OF MULTIMODAL INFILLINGS

We ask human evaluators to judge our synthetic multimodal infillings based on novelty and consistency (§4.3) on a 5-point scale (1-2 = poor, 3 = neutral, 4-5 = good). VCoT infillings outperform all baselines on the 5-point scoring of quality (Figure 6). When comparing win-tie-losses, VCoT also outperforms both baselines by at least 6.6% and 4.4% for the consistency and novelty of our synthetic

Table 2: The downstream wins-tie-loss percentages for VCoT against four baselines for downstream summarization and storytelling tasks. In addition to novelty and consistency, we measure descriptivity and coherence for WIKIHOW and VIST, respectively. We emphasize in purple if VCoT wins or loses by greater than 1%, and we average the scores on the left. VCOT wins or ties in almost every averaged score as well as in most general categories.

| Dataset | VCoT vs. Baselines | Novelty | | | Consistency | | | Descriptivity | | | Average | | |
|---|---|---|---|---|---|---|---|---|---|---|---|---|---|
| | | Win(↑) | Tie | Lose(↓) | Win(↑) | Tie | Lose(↓) | Win(↑) | Tie | Lose(↓) | Win | Tie | Loss |
| WikiHow | Chain-of-Thought | 34.23 | 36.90 | 28.87 | 30.06 | 39.66 | 30.28 | 23.31 | 56.39 | 20.30 | 29.20 | 44.32 | 26.48 |
| | Chain-of-Images | 37.28 | 25.82 | 36.90 | 44.20 | 26.04 | 29.76 | 33.83 | 27.44 | 38.72 | 38.44 | 26.43 | 35.13 |
| | No Infilling | 33.56 | 38.47 | 27.98 | 38.24 | 26.12 | 35.64 | 33.46 | 32.71 | 33.83 | 35.09 | 32.43 | 32.48 |
| | Reference Step | 40.92 | 28.42 | 30.65 | 47.99 | 21.80 | 30.21 | 42.11 | 22.56 | 35.34 | 43.67 | 24.26 | 32.07 |

| Dataset | VCoT vs. Baselines | Novelty | | | Consistency | | | Coherence | | | Average | | |
|---|---|---|---|---|---|---|---|---|---|---|---|---|---|
| | | Win(↑) | Tie | Lose(↓) | Win(↑) | Tie | Lose(↓) | Win(↑) | Tie | Lose(↓) | Win | Tie | Loss |
| VIST | Chain-of-Thought | 39.86 | 29.42 | 30.72 | 30.17 | 38.01 | 31.82 | 35.11 | 35.11 | 29.79 | 35.05 | 34.18 | 30.77 |
| | Chain-of-Images | 33.81 | 23.71 | 42.47 | 35.33 | 30.17 | 34.50 | 35.11 | 36.88 | 28.01 | 34.76 | 30.25 | 34.99 |
| | No Infilling | 34.64 | 23.92 | 41.44 | 37.32 | 27.56 | 35.12 | 31.21 | 43.62 | 25.18 | 34.39 | 34.21 | 31.40 |
| | Reference Step | 28.73 | 40.62 | 30.65 | 37.04 | 30.45 | 32.51 | 36.52 | 38.30 | 25.18 | 34.10 | 36.46 | 29.44 |

| Chain-of-Thought Produced Summary | Visual Chain-of-Thought Produced Summary |
|---|---|
| Monitor your Burmese cat's eating habits and look for signs of diabetes as they age, and take them to the vet regularly for check-ups and preventive care. | Monitor your Burmese cat's eating habits and groom them regularly to check for signs of parasites, infections, or other health problems, and provide a balanced diet with plenty of exercise. |
| Make sure your cat is vaccinated and tested for feline leukemia, and schedule regular check-ups with a veterinarian to ensure your cat is healthy. | Make sure your Burmese cat receives core vaccinations, such as distemper, feline leukemia, and rabies, and consider other vaccinations depending on your cat's needs, as well as spaying or neutering to prevent unwanted litters and reduce the risk of certain diseases. |
| Start deworming your kitten at three weeks old and schedule regular vet visits to monitor their health and ensure they are up-to-date on vaccinations. | Start deworming your Burmese cat at three weeks of age and provide regular flea and tick treatments, vaccinations, and parasite prevention treatments to ensure their health and wellbeing. |

Figure 8: Comparison of a WIKIHOW summary produced by Chain-of-Thought versus Visual Chain-of-Thought. The purple text highlights how VCoT can improve on the summary quality compared to text-only CoT.

visuals *and* text (Table 1), respectively. Qualitative examples show that VCoT outperforms baselines by generating more useful and relevant infillings for sizable logical gaps (Figure 7, Figure 13). The strong consistency and novelty of the synthetic multimodal infillings indicates that they add both relevant and new information to their surrounding sequential steps, and thus VCoT helps bridge logical gaps in sequential data.

VCoT infills chronological gaps in the VIST and WIKIHOW sequences with *consistent* bridging information. We hypothesize the utility of VCoT increases with size of the logical gap because these gaps hinder downstream task performance (§5.4) and interpretability for large language models. VCoT synthetically augments additional context to language models with multimodal infillings, allowing for smaller logical leaps and enhanced reasoning. We argue that VCoT outperforms baselines by maintaining consistency through foveation grounding (§4.2) and CLIP similarity alignment (§4.4).

With regard to novelty, VCoT adds *new, relevant* information for a logical flow between surrounding steps (Figure 11, Figure 12). We observe that many baseline-generated images contain new but less relevant information (Table 1). Notably, COI often generates novel images (Figure 13) that do not align with the surrounding steps, hindering the relevance aspect. In contrast, VCoT excels in increasing yet balancing both newness and relevance.

## 5.4 DOWNSTREAM TASK PERFORMANCE

**Qualitative Results** It is clear from our qualitative results (Figure 8, Figure 9) that VCoT increases consistency among the text and images, while also adding relevant, novel information. Additionally, the infillings provide multimodal interpretability into computer reasoning (Figure 4).

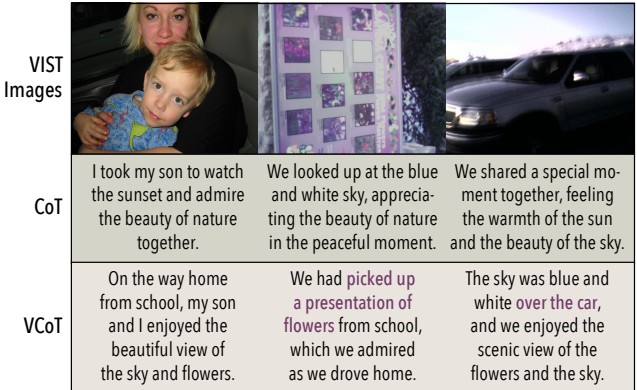

Figure 9: Comparison of a VIST story produced by CoT versus Visual Chain-of-Thought. The purple text highlights how VCoT can improve on the storytelling quality compared to text-only CoT.

**Quantitative Results**  VCoT convincingly surpasses baselines in average downstream results (Table 2), winning in every category besides tying Chain-of-Images for VIST.

For WIKIHOW, we see a clear improvement in novelty (Table 2), which makes sense because we augment the data with novel, relevant tokens for instruction generation. Meanwhile for consistency, VCoT ties with CoT and surpasses all other baselines. We hypothesize this tie is due to the lengthy nature of the input text, allowing GPT-3.5 to create fairly consistent instructions with or without the use of VCoT. For descriptiveness, VCoT beats all baselines except for CoI, likely because CoI injects visually-descriptive information that isn't grounded by the original text. Specifically, CoI's image-only modality allows it to capture novel but less consistent information than that of the limited textual description. Overall, VCoT offers a balanced combination of performance improvements.

For VIST, VCoT's infillings improve consistency against all baselines besides CoT. Conversely, we find that VCoT loses in novelty to all baselines other than CoT, which VCoT (Table 2) improves upon. These juxtaposed results offer interpretability into computer reasoning, namely that CoI introduces more novel information, CoT preserves consistency, and VCoT maintains a balance. We suspect that VCoT's loss in novelty is a result of 1) VCoT's attentiveness to consistency through CLIP guidance and multipoint foveation, and 2) repeated tokens generated with our consistency-driven approach that overlap and cause repetition. These results demonstrate that VCoT provides insight on specific ways to improve computer reasoning–design strategies that both enhance and balance novelty and consistency.

## 6   CONCLUSION

We introduce a new research direction to generate synthetic logical infillings for sequential data, which we tackle with our novel multimodal paradigm visual chain-of-thought. We combine chain-of-thought with visual guidance to recursively generate multimodal infillings that bridge the natural logical gaps between sequential elements. By adding infillings to sequences while maintaining consistency, we augment novel, relevant information to bolster downstream task performance while also providing human-interpretable insights into the system's reasoning process. Through task unification, we can apply VCoT on various multimodal tasks.

Human experiments show that VCoT creates more novel and consistent logical infillings than the uni-modal CoT and CoI baselines performed in parallel on the sequential datasets VIST and WIKIHOW, and these infillings are helpful to improve downstream task performance. While we demonstrate VCoT on the instruction summarization and visual storytelling tasks, future work can explore VCoT in new domains that could benefit from synthetic data augmentation and bolstered reasoning abilities, such as procedural planning, DNA sequencing, and video understanding. Furthermore, future research can look into aligning multimodal infillings with other desired downstream performance metrics. Along these lines, it is valuable to measure desired outputs through automatic evaluation metrics to support evaluation at scale.

ETHICS STATEMENT

Human evaluation was conducted via crowdsourcing using the Amazon Mechanical Turk platform. For all Mechanical Turk experiments, we require workers to be located in Australia, Canada, New Zealand, the United Kingdom, or the United States and have a HIT approval rating of at least 98%. For both the intermediate and downstream tasks, it is estimated we pay workers at a rate of $15.0/hr, a rate which is above the federal minimum wage at the time in which this research was conducted. The data annotation project is considered an exempt protocol by the human subject committee by IRB guidelines. We provide a consent form at the beginning of each task. We include additional details regarding screenshots and task descriptions for each Mechanical Turk study in Appendix A.

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

# A APPENDIX

## A.1 ALGORITHMS

---

**Algorithm 2:** recGen(prev, next, depth, f)

---

**Data:** (image, text) pairs prev $(v_1, t_1)$ and next $(v_2, t_2)$, depth $d$, and foveation $f$
**Result:** infilled list of (visual, text) pairs between $(v_1, t_1)$ and $(v_2, t_2)$

1   $infilling \leftarrow$ genInfilling$(prev, next, f)$
2   **if** $d + 1 \leq$ *depth-limit* **then**
3     $prevGen \leftarrow recGen(prev, infilling, d + 1, f)$
4     $nextGen \leftarrow recGen(infilling, next, d + 1, f)$
5     **return** $prevGen + [infilling] + nextGen$
6   **return** $[infilling]$

---

VCoT's recursive multimodal infilling generation algorithm given two sequential text-visual pairs.

## A.2 CROWDSOURCING HUMAN EVALUATION

We show the interface of our human evaluations in Figure 10. We manually ensure no personal information is collected and no offensive content is presented during human evaluations.

## A.3 DETAILS OF OUR APPROACH

We apply our generated multimodal infillings to improve the traditionally text-based tasks of visual storytelling and instruction summarization. To integrate our multimodal infillings into the downstream tasks, we pass them along with the unified input to create an extensive summary using few-shot examples and image captioning. The information in the summary can guide models to better reason temporally in a wide range downstream tasks.

Because storytelling and summarization are inherently different, we use different prompting schemes. For VIST, we pass in the few-shot examples, summary, focus, current steps, and past story steps to autoregressively build the final sequential story. Since it is very important for a story to flow over time, current steps are very dependent on the past steps in time. For Wikihow, we also pass in the few-shot examples, summary, focus, current steps. Unlike the past story steps for VIST, we input the surrounding multimodal infillings because the summarized steps of WIKIHOW articles aren't quite as dependent on flowing over time. Summarizing "How-To" articles into a series of human-understandable instructions does, however, require understanding the nearby logical steps.

Experiments demonstrate that VCoT improves the overall quality of the final stories and instruction summarization for the example datasets WIKIHOW and VIST.

### A.3.1 LIKELIHOOD OF GPT-3.5 OUTPUTS

The GPT-3.5 API response[4] returns the log probabilities of individual generated tokens. We compute the joint log likelihood probability of an output sequence $t_1, ..., t_n$ as the sum of the individual token log probabilities (Equation 8).

$$\ln(\mathbb{P}(t_1, ..., t_n)) \approx \sum_{i=1}^{n} \ln(\mathbb{P}(t_i)) \tag{8}$$

## A.4 REPRODUCIBILITY

The only non-open source model we use is GPT-3.5 `text-davinci-003`. We used the OpenAI API in January 2023 with temperature 0 for one candidate infilling and temperature 0.5 to generate 4 different candidate infillings, and temperature 0 otherwise for all other tasks; all other hyperparameters are set to the default. Repetitive runs on VIST/WIKIHOW examples yield similar results, promoting reproducibility.

---

[4]https://platform.openai.com/docs/api-reference/completions/create#completions/create-logprobs.

## Instructions

You will be given a series of (text, image) pairs in sequence. Then, you will be given intermediate (text, image) pairs that try to fill in the logical gaps between the original sequence. You will be asked to rate the **consistency** and **novelty** of these intermediate (middle) pairs with respect to the original (surrounding) sequence.

Note that the intermediate pairs are signified by a red box.

A text/image pair is **consistent** if it's information comes from and references the surrounding pairs in the sequence.
A text and image is **novel** if it adds relevant and new information that is not already in the immediate previous or next step.

- Any incoherent responses or extremely poor responses will be rejected.
- Strong responses will get invited for follow-up tasks.
- Do not complete this task more than once.

**Example 1:**

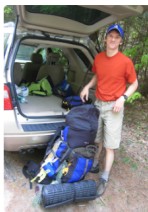 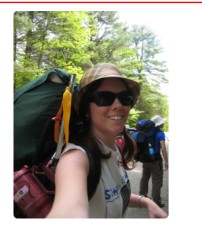 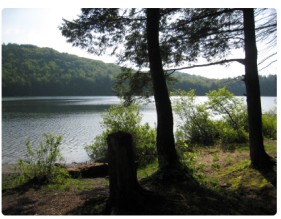

A young man is getting ready to leave for a camping trip by packing up all of his supplies in a car.

Having unpacked the car, the smiling young woman with a camping backpack walks behind the man, carrying supplies into the forest.

A large forest surrounds a campground with a lake surrounded by vegetation.

Text Consistency: 5. The middle text is consistent with many details (the camping trip, packing/unpacking a car, carrying supplies, and a man) from the surrounding pairs. So, it is "very good".

Image Consistency: 4. The middle image is consistent with most details (the man, forest, supplies, excited faces) from the surrounding pairs so it's "good". To improve it could includ information about the car or driving to the forest.

Text Novelty: 4. The middle text introduces relevant information about a young woman going on a camping trip and walking with the man, so it's "good". To improve it could indicate that they're walking to the lake/campsite.

Image Novelty: 5. The middle image introduces relevant information about the young woman character as well as some camping supplies like a backpack and sunhat. So, it is "very good".

Focus: The focus of the left and right image/text pairs is a man going camping in the woods.

## Rate the consistency and novelty of each of the scenarios.

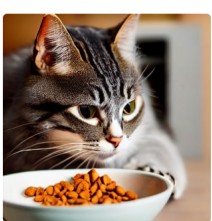 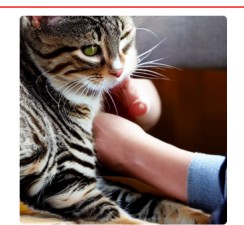 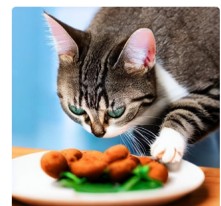

To encourage your cat to obey when offered food, pick up any remaining food in its bowl after each feeding and use it for teaching tricks.

Train your cat by repeating commands and providing gentle guidance when necessary to help it learn the desired behavior.

Reward your cat with the food after it completes a task correctly to reinforce the desired behavior.

Rate the **consistency of the middle image** with respect to the left and right pairs.  ○ 1 (very poor) ○ 2 (poor) ○ 3 (neutral) ○ 4 (good) ○ 5 (very good)

Rate the **consistency of the middle text** with respect to the left and right pairs.  ○ 1     ○ 2     ○ 3     ○ 4     ○ 5

Rate the **novelty of the middle image** with respect to the left and right pairs.  ○ 1     ○ 2     ○ 3     ○ 4     ○ 5

Rate the **novelty of the middle text** with respect to the left and right pairs.  ○ 1     ○ 2     ○ 3     ○ 4     ○ 5

Figure 10: Amazon Mechanical Turk Platform Interface

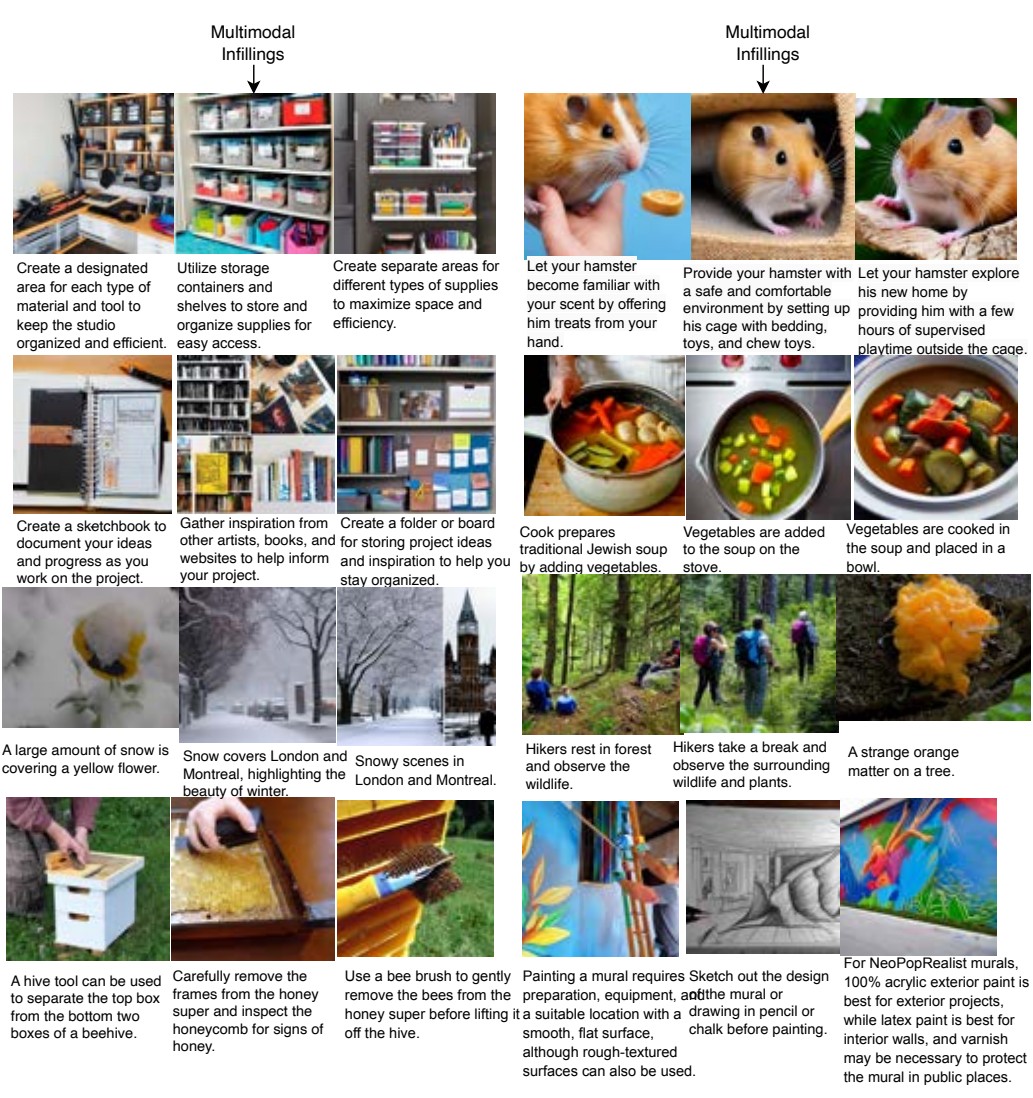

Figure 11: Example VCoT multimodal infillings (middle text-visual pair) generated with our visual chain-of-thought method.

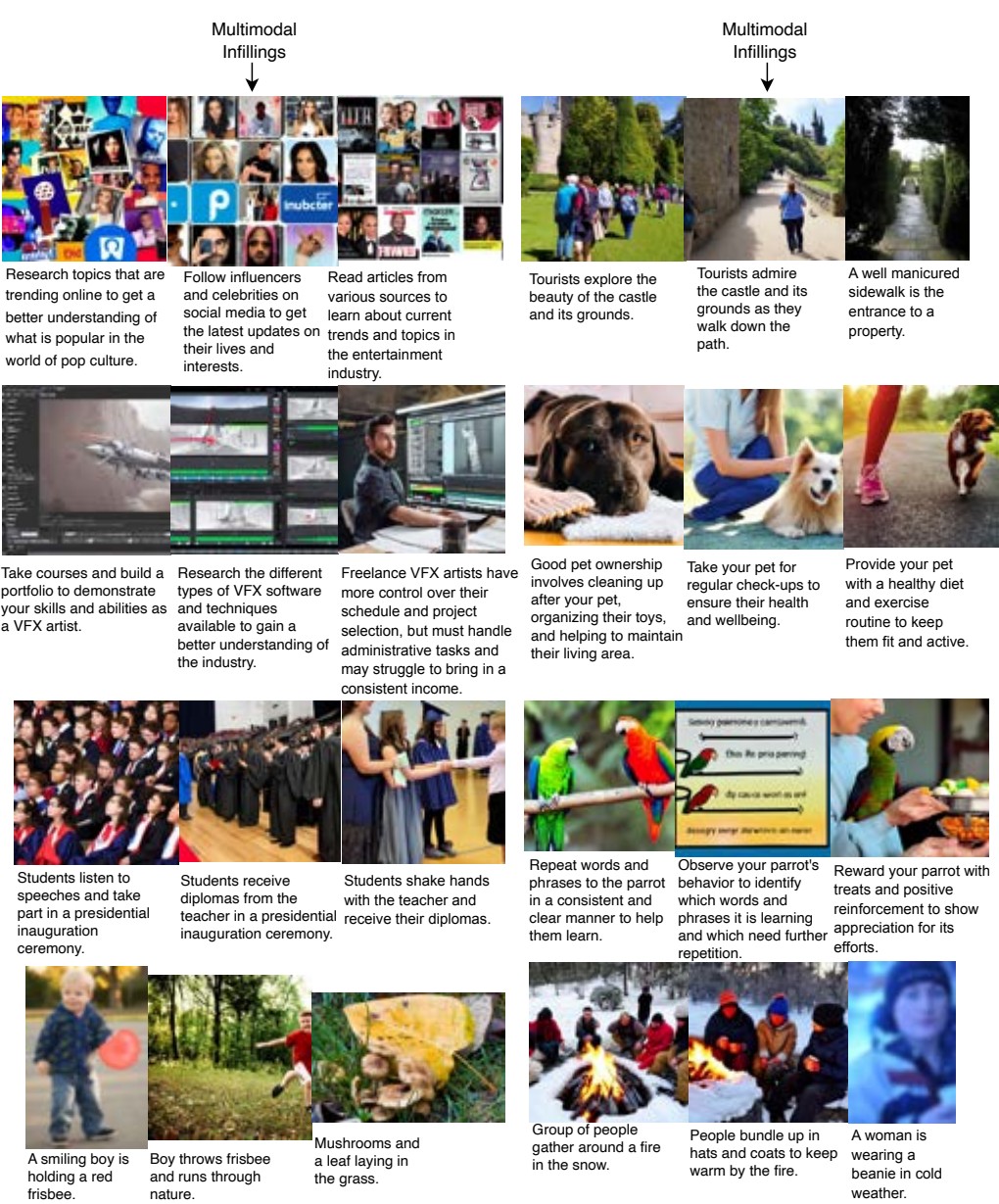

Figure 12: Example VCoT multimodal infillings (middle text-visual pair) generated with our visual chain-of-thought method.

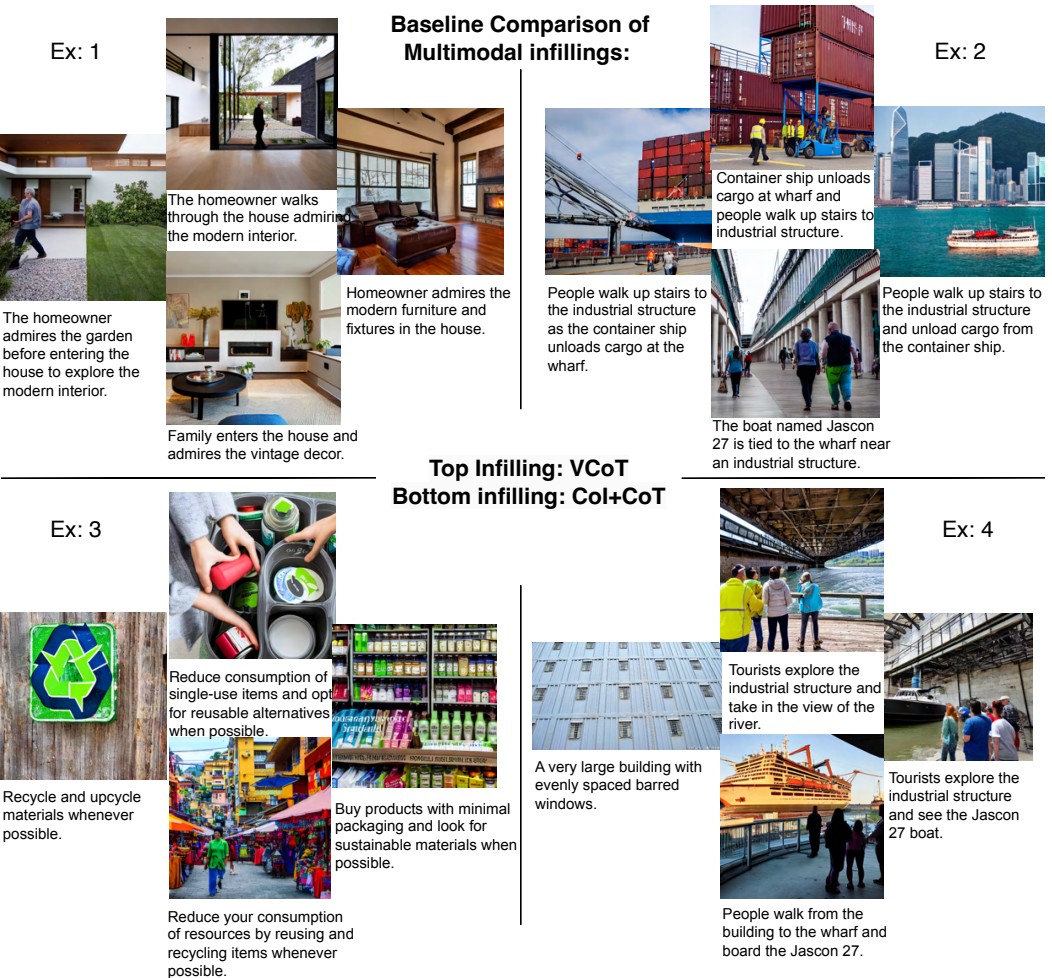

Figure 13: Comparison of generating multimodal infillings for two surrounding steps using visual chain-of-thought vs text-only chain-of-thought plus image-only chain-of-images performed in parallel.

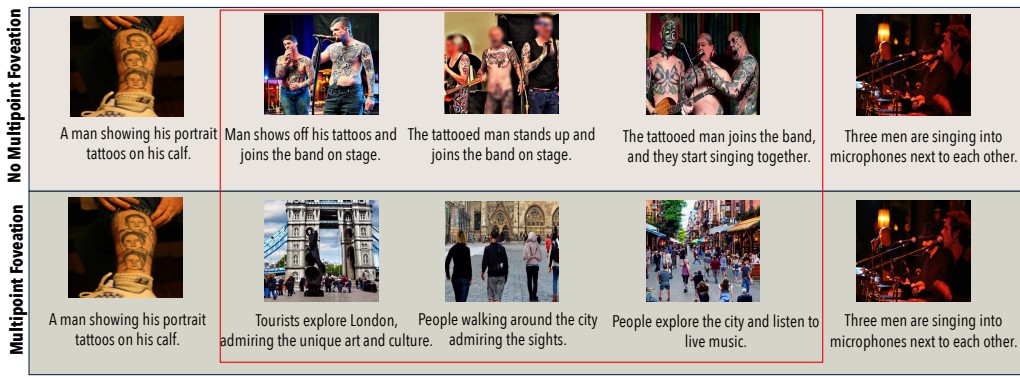

Figure 14: Comparison of infillings generated with and without Multipoint Foveation (MPF) in the VIST dataset. In this example, the overall story is about people exploring different parts of a city, including a tattoo parlor, music show, city streeets, etc. The infilling generated with MPF is more consistent with the global context of the story. By contrast, the infilling generated without MPF overfits to the local information of the man with the tattoo and creates an unrealistic tattooed singing man inconsistent with the actual story.

