# OpenReview forum: "Visual Chain of Thought: Bridging Logical Gaps with Multimodal Infillings"
_ICLR.cc/2024/Conference — ICLR 2024 Conference Withdrawn Submission_

### Official Review · Reviewer_E16R · 2023-10-24

**Soundness:** 2 fair
**Presentation:** 2 fair
**Contribution:** 2 fair
**Rating:** 3
**Confidence:** 4

**Summary:**

This manuscript presents Visual Chain-of-Thought (VCoT) which extends chain-of-thought prompting (CoT) in NLP literature to vision-and-language tasks. The paper considers the situation where the sequential data contains logical gaps. The core idea is to fill in the gaps by generating synthetic vision-and-language data (i.e., multimodal infillings).

VCoT first extracts the global information of the sequential data, multipoint foveation (MPF), by using several pre-trained models, such as StableDiffusion, CLIP, and GPT-3.5. Then, the proposed approach recursively generates the multimodal infillings (i.e., text-visual pairs) based on the original sequential data and MPF. In each step of generation, VCoT selects the most consistent multimodal infillings among candidates after generating multiple candidates. Such a generation process is repeated a fixed number of times.

The paper conducts experiments regarding the quality of the generated multimodal infillings through human evaluation and visualization.

**Strengths:**

(S1) Generating intermediate data to bridge the logical gaps is definitely a novel direction.

(S2) The work conducts a human evaluation to assess the generated infilling’s quality.

**Weaknesses:**

(W1) The generated multimodal infillings show marginal improvements compared with the simple baselines (Chain-of-Thought and Chain-of-Images). As can be seen in Table 2, VCoT has a roughly 30% chance of winning on average. It indicates that the proposed method does not dominate the baselines in downstream tasks (i.e., WikiHow and Visual Storytelling). Even worse, the gap between VCoT and the method that does not use any infillings (No Infilling) also seems marginal, although VCoT leverages several strong pre-trained models (e.g., GPT-3.5, CLIP, StableDiffusion, and OFA) with massive computation.


(W2) The paper mainly focused on checking the quality of the generated multimodal infillings. It would be great if the paper presented the impact of VCoT in existing evaluation protocols at VIST and WikiHow tasks.


(W3) There are several components in VCoT, but the paper does not present any ablation studies. So, I can’t check the individual contribution of the proposed components. For example,
- VCoT generates multipoint foveation (MPF) only with text (i.e., w/o image captions)
- VCoT generates multimodal infillings without MPF (i.e., w/o MPF)
- VCoT generates a single multimodal infilling, not multiple (i.e., five) candidates (i.e., w/o consistency-driven selector)


(W4) There are too many typos and inconsistent notations in the manuscript as below. I request the authors to review the entire paper thoroughly.
- In Section 3, the main text describes that the judgment function j takes vision and language data, but the function takes the text in Eq. 2.
- In Section 3, textual references of Equation 3 and 4 should be Equation 5 and 6, respectively.
- In Equation 6, “Eq. 6 holds” should be “Eq. 5 holds”
- In Section 4, textual references of Equation 5 should be Equation 7.

**Questions:**

Please see the weaknesses

---

### Official Review · Reviewer_BRKH · 2023-10-31

**Soundness:** 3 good
**Presentation:** 3 good
**Contribution:** 3 good
**Rating:** 6
**Confidence:** 4

**Summary:**

This paper addresses the problem of visual chain-of-thought, which is formulated as an infilling task that constructs intermediate states based on information from their adjacent states. It takes advantage of a large language model (GPT 3.5) and a diffusion model to generate textual descriptions and images, and adaptively selects candidate text/image by evaluating their novelty and consistency w.r.t. adjacent states. Experimental results show that the proposed approach is able to outperform single-modal chain-of-thought methods.

**Strengths:**

+ This paper addresses an important problem of extending chain-of-thought logical reasoning from single-modal to multi-modal domain.

+ The method takes into account the relationship between the intermediate states and neighbors (i.e., novelty and consistency), showing enhanced performance in generating coherent text and images.

+ Large-scale human evaluation demonstrates the advantages of the proposed approach.

**Weaknesses:**

- There is a lack of ablation study on the proposed method, making it difficult to estimate the effectiveness of individual components. For instance, while the recursive generation seems to be an important step, the paper does not provide any quantitative/qualitative evaluation of how such a process is advantageous to its counterpart (e.g., iterative). Looking at Figure 7, the text generated by the proposed method does not appear to be very novel compared to the one for the baseline.

- While it could be unfair to directly compare the proposed zero-shot method with supervised approaches, it would still be reasonable to conduct evaluation on the ground truth annotations. Otherwise, it is unclear whether or not the method truly formulates the intermediate logical procedure for problem-solving.

- Despite the superiority of the proposed method reported in Table 1, it appears that there is no clear advantage when considering different approaches independently (i.e., Figure 6).

- The paper only considers a simplified setting of chain-of-thought, where the goal is to generate a single intermediate step based on its two adjacent neighbors. On the other hand, for complicated problems, multiple logical steps are typically required to form the chain-of-thought. How would the proposed approach generalize to these scenarios? More importantly, it is also unclear whether or not it can address out-of-domain reasoning problems, considering its heavy reliance on a pretrained large language model for inferring the intermediate steps.

**Questions:**

(1) What is the performance of the method when evaluating with ground truth annotations?

(2) Can the method handle a longer chain-of-thought?

(3) Please provide additional results to support the effectiveness of the recursive algorithm.

(4) Does the method work on visual reasoning tasks that have a stronger emphasis on logical reasoning?

---

### Official Review · Reviewer_a5oz · 2023-10-31

**Soundness:** 2 fair
**Presentation:** 3 good
**Contribution:** 2 fair
**Rating:** 3
**Confidence:** 4

**Summary:**

This paper proposed a method named visual chain-of-thought to infill the logical gaps inside multimodal sequential data. Specifically, this method first generates a global text focus and then recursively generates infilling texts and images by prompting ChatGPT and Stable Diffusion. In experiments, they evaluate their method by humans and show it can outperform the baselines.

**Strengths:**

1. The idea of visual chain-of-thought matches the task of multimodal infilling well.
2. The paper includes a lot of qualitative results and human evaluations against the baseline, which looks promising.

**Weaknesses:**

1. Lack of verification on downstream tasks. Although the authors mention a lot about downstream evaluation in the paper, it mainly focuses on directly evaluating the quality of the generated image/text in the downstream dataset.s However, a more important dimension, IMO, is whether the infilled image/caption is useful for downstream tasks. For example, when adding the generated data as additional training data for visual storytelling and wiki-how, if the performance of the SOTA model  (retrieval/generation) can be further boosted or not?
2. About the recursive approach to generate the infillings. It's claimed in the paper a lot of advantages of `recursive approach` vs `iterative approach` about dynamics, such as `A recursive approach makes it easier to dynamically determine when infillings are not beneficial to the task and when to stop generating them in contrast to an iterative approach`. However, in Sec. 4.4, the authors admit that a fixed depth is the best in practice, which eliminates the advantages claimed before, especially about dynamically determining/stopping.
3. I don't quite get why the recursive approach is called `novelty-driven`. To me, there is nothing emphasizing adding additional novel content in the recursive algorithm.
4. There are many hyperparameters in the paper, such as the number of text candidates, number of image candidates, number of few-shot examples. However, no ablation experiments are conducted to validate them.
5. No ablation on the effectiveness of core components of the method, eg, multipoint foveation, novelty/consistency-driven infilling.
6. How are the generated multimodal infillings compared with ground-truth quality? It's the upper-bound of this task and should be evaluated too.

**Questions:**

1. The details of COT and COI parallel baseline are missing.
2. What does the prompt of multipoint foveation look like?

---

### Official Review · Reviewer_dqPG · 2023-11-04

**Soundness:** 2 fair
**Presentation:** 2 fair
**Contribution:** 2 fair
**Rating:** 5
**Confidence:** 4

**Summary:**

The paper extends the paradigm of CoT from reasoning in language-only QA tasks to multi-modal (vision-language) data tasks that are complex and involve creativity.

The proposed method VCoT (Visual Chain-of-Thought) introduces synthetic infillings in temporal sequences that aid in solving downstream tasks. The main goal is to fill “reasoning gaps” that exist in temporal sequences.

VCoT is evaluated on two datasets — Visual Storytelling (Vist) and WikiHow. Humans find that VCoT outperforms text-only-CoT and vision-only-CoT baselines.

**Strengths:**

Chain-of-Thought based reasoning is now a well-established technique in Question-Answering and reasoning tasks in NLP. The proposed novel multi-modal Visual Chain-of-thought approach demonstrates improved performance on variations of downstream storytelling (VIST) and sequential reasoning task (WikiHow) tasks / datasets. The overall results are evaluated via human studies along multiple relevant dimensions of image and text qualities.

The two main constraints of novelty and consistency are reasonable — it prevents redundancy and ensures high relatedness of the generated image with the other images in the sequence.

Images provide additional information that is absent in the textual modality. The synthetically generated Chain-of-Thought text and corresponding images serve as a different source of creative details for downstream tasks like storytelling.

**Weaknesses:**

One of the paper’s major motivations for VCoT is that it provides a human-interpretable insight into the AI system’s reasoning ability. However it’s not clear in what way, and which particular model’s predictions are explained by the proposed, synthetically generated VCoT images.

Some details of the approach are unclear from the paper.

- E.g., in Eq. (2), the argmax is computed over CLIP similarity scores between the current predicted text t_i, and what other element?
- The notation in the Equation series (1-5) is also confusing. g appears to be a generative model that predicts text t_i in Eq. (1), and image v_i in Eq. (3). However, an argmax is computed over g(.) in Eq. (4), as if it were a score function.
- What exactly is the scoring function, using which is  v_i^{best} chosen in Eq. (4) ?
- It appears that the Equations being referenced in the text of Sec. 3 and Sec. 4.2 might be incorrect.

Unfortunately, the current version of Fig. 2 doesn’t clarify things either. The different blocks in the current color-scheme appear too similar to each other, and distinguishing between them is difficult for this reader.

During task unification (of text-only WikiHow data), individual images are generated given individual text instructions. A number of candidate images for the time-step i are generated, and CLIP embedding similarity between the images and the corresponding text t_i is computed. The most relevant image is chosen. While this provides a sequence of images, there is no constraint that the images themselves, are “coherent” (related to each other). This could result in a sequence of different, apparently unrelated images despite the text being coherent and related to the corresponding images.

It would be good if it were possible to scale the evaluation to a larger dataset. Currently, evaluation is only based on human feedback. However, this approach only scales at a significant cost. If the authors present an automated method ot evaluate the different aspects of the model, that might be a good contribution that makes the paper stronger.

The overall approach is rather involved, with multiple blocks / processes that generate the final result. However, the influence of the individual blocks, e.g., “multi-point foveation”, “task unification” (+ captioning), etc., are not evaluated or discussed in detail.


Minor comments:

- Fig. 3 (left) typo -- (v_i’’, t_i’’) --> (v_i', t_i')?
- The colors of the different blocks in the architecture diagram in Fig. 2 are too similar to each other.

**Questions:**

1. Are there any ablation studies regarding the influence of the different parts of the overall system?

2. Are there any possible automatic metrics that can be used to (at least approximately) evaluate performance of VCoT?

3. Algorithm 1 seems to describe details wrt "consistency". Could you please also explain in detail, the algorithm design that ensures "novelty"?

4. A key advantage of VCoT is to reduce logical gaps that are present, e.g., in VIST samples. Is there ny way to quantify the semantic concept of "logical gap" with a metric? If automatic metrics are insufficient, then even a metric based on (multiple, averaged) human scores might be useful.